# ClickDiff: Click to Induce Semantic Contact Map for Controllable Grasp Generation with Diffusion Models

Peiming Li*
State Key Laboratory of General Artificial Intelligence, Peking University, Shenzhen Graduate School
Shenzhen, China
lipeiming1001@stu.pku.edu.cn

Ziyi Wang*
State Key Laboratory of General Artificial Intelligence, Peking University, Shenzhen Graduate School
Shenzhen, China
ziyiwang@stu.pku.edu.cn

Mengyuan Liu†
State Key Laboratory of General Artificial Intelligence, Peking University, Shenzhen Graduate School
Shenzhen, China
nkliuyifang@gmail.com

Hong Liu
State Key Laboratory of General Artificial Intelligence, Peking University, Shenzhen Graduate School
Shenzhen, China
hongliu@pku.edu.cn

Chen Chen
Center for Research in Computer Vision, University of Central Florida
Orlando, USA
chen.chen@crcv.ucf.edu

## ABSTRACT

Grasp generation aims to create complex hand-object interactions with a specified object. While traditional approaches for hand generation have primarily focused on visibility and diversity under scene constraints, they tend to overlook the fine-grained hand-object interactions such as contacts, resulting in inaccurate and undesired grasps. To address these challenges, we propose a controllable grasp generation task and introduce ClickDiff, a controllable conditional generation model that leverages a fine-grained Semantic Contact Map (SCM). Particularly when synthesizing interactive grasps, the method enables the precise control of grasp synthesis through either user-specified or algorithmically predicted Semantic Contact Map. Specifically, to optimally utilize contact supervision constraints and to accurately model the complex physical structure of hands, we propose a Dual Generation Framework. Within this framework, the Semantic Conditional Module generates reasonable contact maps based on fine-grained contact information, while the Contact Conditional Module utilizes contact maps alongside object point clouds to generate realistic grasps. We evaluate the evaluation criteria applicable to controllable grasp generation. Both unimanual and bimanual generation experiments on GRAB and ARCTIC datasets verify the validity of our proposed method, demonstrating the efficacy and robustness of ClickDiff, even with previously unseen objects. Our code is available at https://github.com/adventurer-w/ClickDiff.

*Both authors contributed equally to this research.
†Corresponding author is Mengyuan Liu (e-mail: nkliuyifang@gmail.com).

## CCS CONCEPTS

• **Computing methodologies → Shape inference**.

## KEYWORDS

Controllable Grasp Generation, Human-Object Interaction, Semantic Contact Map

**ACM Reference Format:**
Peiming Li, Ziyi Wang, Mengyuan Liu, Hong Liu, and Chen Chen. 2024. ClickDiff: Click to Induce Semantic Contact Map for Controllable Grasp Generation with Diffusion Models. In *Proceedings of the 32nd ACM International Conference on Multimedia (MM '24), October 28-November 1, 2024, Melbourne, VIC, AustraliaProceedings of the 32nd ACM International Conference on Multimedia (MM'24), October 28-November 1, 2024, Melbourne, Australia.* ACM, New York, NY, USA, 9 pages. https://doi.org/10.1145/3664647.3680597

## 1 INTRODUCTION

In recent years, the modeling of hand-object interactions [1, 2, 5, 7, 11, 15, 24, 25, 30, 34, 38] has gained substantial importance due to its significant role in applications across human-computer interaction [39], virtual reality [13, 39, 40], robotics [3, 37, 45], and animation [27]. Addressing the specific needs for hand-object interaction modeling emerges as a paramount concern. How do hands interact with a bowl? One can envision a variety of interaction types (*e.g.*, "grabbing/holding") and numerous possible interaction locations (*e.g.*, "rim/bottom"). The diversity of these interactions largely stems from the layouts of hands and objects, making the generation of accurate interactions challenging. An accurate generative model should account for factors such as which areas of the object will be touched and which parts of the hand will make contact. In contrast, a lack of thorough and precise modeling may result in unnatural and unrealistic interactions.

Traditional approaches [15, 24] for hand generation have primarily focused on visibility and diversity under scene constraints. Most existing controllably generated hand-object interaction images rely on simple textual scene understandings, such as "a hand holding a cup" for input synthesis, failing to generate precise hand-object interactions due to the lack of fine-grained information. Methods

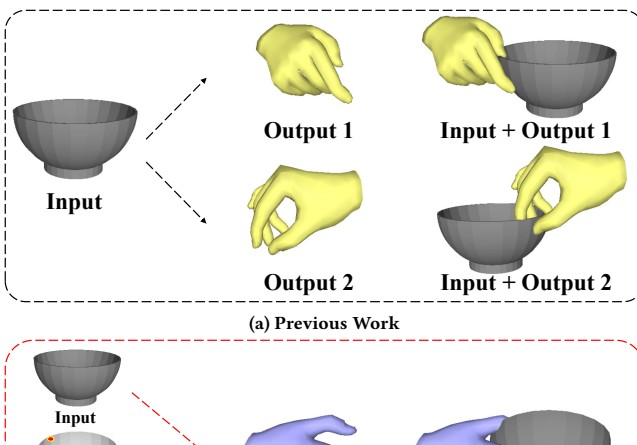

**(a) Previous Work**

**(b) Controllable Grasp with SCM (Ours)**

**Figure 1: Previous works face issues of contact ambiguity, where an input object could lead to multiple undesired grasps, such as when a bowl is expected to be grabbed from the bottom, revealing the importance of controllable grasp generation. By manually determining contact points and specifying contact fingers, one can obtain SCM (each color represents the contact area of a different finger) by traversing the area around the clicked point. Finally, by utilizing SCM, it's possible to achieve accurate user-expected grasp.**

[44] utilizing contact information often depend on contact mapping applied to object point clouds to indicate contact points. However, relying solely on information learned from contact maps as constraints still leaves the issue of which areas of the hand to use for touching unaddressed. Moreover, providing contact maps for both the object and the hand simultaneously is almost an impossible task in real-world applications.

Previous efforts in grasp generation have emphasized achieving visibility and diversity, often overlooking the fine-grained hand-object interactions such as contacts. This oversight has led to a notable deficit in both accuracy and practicality of generated grips, especially in scenarios requiring precise and controllable synthesis. To address these challenges, we propose a controllable grasp generation task, which can realize the generation through manually specified hand and object contact point pairs. Our key insight is that the movements of hand joints are largely driven by the geometry of fingers. Explicitly modeling the contact between fingers and object sampling points during training and inference can serve as a powerful proxy guide. Employing this guidance to link the movements of hands and objects can result in more realistic and precise interactions. What's more, grip generation guided by artificially defined contact is more practical.

Fig. 1 demonstrates the contact ambiguity issues present in past generation methods compared to controllable generation task with artificially designated contact points. To better model the artificially defined fine-grained contacts shown in Fig. 1, we introduce a novel, simple and easily specified contact representation method called Semantic Contact Map. By processing point clouds of objects and hands, it obtains representations of points on the object that are touched and the fingers touching those points, specifically providing: (1) The points on the object that are touched. (2) The number of the finger touching the point. A significant advantage is that one can click to customize the Semantic Contact Map to achieve user-controlled interactive generation. We show that utilizing Semantic Contact Map can achieve more natural and precise generation results than conventional methods that rely solely on contact map. Furthermore, to better utilize a human-specified SCM in contact supervision constraints, we propose a Dual Generation Framework based on conditional diffusion model [12, 19, 36, 43]. The denoising process of the diffusion model inherently involves constraints on the parameters of both hands, reducing unnatural and unrealistic interactions between them. Building on the aforementioned observations, this paper introduces the ClickDiff based on Semantic Contact Map to realize controllable grasp generation, which is divided into two parts: (1) We explicitly model the contact between fingers and object sampling points by specifying the locations of contact points on the object and the finger numbers contacting those points, guiding the interactions between hands and objects. Benefit from previous work utilizing contact map [6, 10, 15, 24, 33], to embed Semantic Contact Map into the feature space for easy learning and utilization by the network, we propose the Semantic Conditional Module, which can generate plausible contact maps based on specified Semantic Contact Map and object point clouds. (2) Inspired by the grasps generation method in the RGB image domain [10], we propose the Contact Conditional Module, which synthesizes hand grasps based on generated contact maps and object point cloud information as conditions. To better utilize Semantic Contact Map, we propose Tactile-Guided Constraint. Tactile-Guided Constraint can extract pairs in the SCM that represents touching and integrate contact information into the Contact Conditional Module by calculating the distance between the centroid of each finger's predefined set of points and the contact point on the object. Specifically, we provide a Dual Generation Framework that allows using Semantic Contact Map on a given object as a condition for generating contact maps, which then serve as a condition for generating grasps.

In summary, our contributions are as follows:

- We are the first to propose the controllable grasp generation task. We introduce a new contact representation method, named Semantic Contact Map. Our method enables more precise generation through user-specified or algorithmically predicted Semantic Contact Map.
- We propose a Dual Generation Framework composed of Semantic Conditional Module and Contact Conditional Module. The former generates contact maps using Semantic Contact Map as conditions. The latter uitilizes Tactile-Guided Constraint, addressing the contact ambiguity issue in direct contact map generation.
- We evaluate the evaluation criteria applicable to controllable grasp generation. Our approach outperforms general grasp generation baselines on the GRAB and ARCTIC datasets.

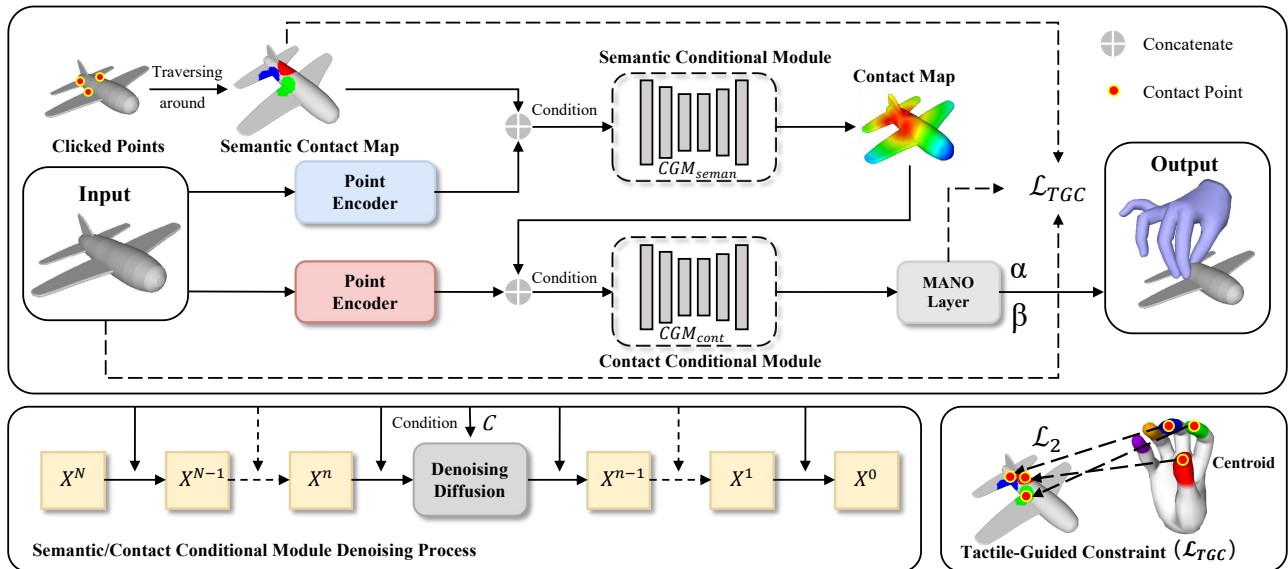

**Figure 2: Overview of ClickDiff: The model initially takes an object's point cloud as input and predicts the contact map conditioned on the Semantic Contact Map within the Semantic Conditional Module. Subsequently, the predicted contact map is fed into the Contact Conditional Module, where grasping is generated under the guidance of TGC and contact map.**

Additionally, we verify its robustness for unseen and out-of-domain objects.

## 2  RELATED WORK

### 2.1  Controllable Human-Object Interaction

In recent years, there have been attempts to model fine-grained interactions between the human and objects. Recent methods have shifted focus towards the prediction of static human poses that are congruent with environmental constraints, particularly in scenarios involving affordances and hand-object interactions. For instance, COUCH [44] presents a comprehensive model and dataset designed to facilitate the synthesis of controllable, contact-based interactions between humans and chairs. Furthermore, there has been notable exploration into the generation of motion conditional on external stimuli, such as music [23]. TOHO [22] exemplifies this by generating realistic and continuous task-oriented human-object interaction motions through prompt-based mechanisms. Additionally, advancements in custom diffusion guidance have significantly enhanced both the controllability [14, 16, 28] and physical plausibility [41] of generated interactions. EgoEgo [21] fed head pose to a conditional diffusion model to generate the full-body pose. Nevertheless, a gap remains in concerning methods that specifically address fine-grained interactions, particularly the nuanced contacts involved in hand-object interactions. Inspired by the aforementioned methods, our method enables a more precise prediction and control of grasping through user-specified or algorithmically predicted contacts.

### 2.2  Grasp Generation

The task of generating object grasps has undergone significant evolution, greatly benefitting from the introduction of novel 3D datasets. Taheri *et al.* [33] notably extended this research through the development of the GRAB dataset, which not only delineates

the hand's contact map but also incorporates the entire human body's interaction with objects. ARCTIC [9] introduces a dataset of two hands that dexterously manipulate objects. It contains bimanual articulation of objects such as scissors or laptops, where hand poses and object states evolve jointly in time. To ensure the generated grasps with both physical plausibility and diversity, the majority of existing models employ a Conditional Variational Autoencoder (CVAE) framework to sample hand MANO parameters [15, 29, 32, 33, 35] or hand joints [17], thus modeling the grasp variability primarily within the hand's parameter space. Given an input object, liu *et al.* propose a novel approach utilizing a conditional generative model based on CVAE, named ContactGen [24], which, coupled with model-based optimization, predicts diverse and geometrically feasible grasps. Meanwhile, GraspTTA [15] introduces an innovative self-supervised task leveraging consistency constraints, allowing for the dynamic adjustment of the generation model even during testing time. Despite these advances, the CVAE models often overfit to prevalent grasp patterns due to the human hand's high degree of freedom, leading to the production of unrealistic contacts and shapes. To address these challenges, our work employs a conditional diffusion model framework, which uniquely focuses on improving the fidelity of generation through user-specified contacts.

## 3  OVERVIEW

In this work, we primarily aim to answer two questions: (1) How can we characterize fine-grained contact information simply and efficiently to achieve controllable grasp generation? (2) How can we better utilize the related contact information to guide grasp generation? To address question (1), in Sec. 4.1, we propose a controllable contact representation Semantic Contact Map (SCM), which simultaneously represents the fine-grained contacts between fingers and

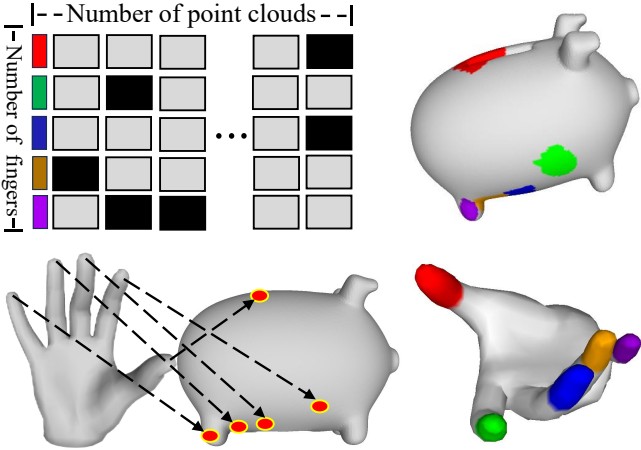

**Figure 3: Illustration of the Semantic Contact Map. The fingers are divided into five parts, represented by different colors. The SCM indicates the points on the object that are being touched and the finger parts touching these points. Each point may be touched by more than one finger.**

objects. In Sec. 4.2 and Sec. 4.3, we propose a Dual Generation Framework and Tactile-Guided Constraints (TGC), using SCM to solve the contact ambiguity problem, thereby addressing question (2). Fig. 2 summarizes our process for generating grasps.

# 4 METHOD

## 4.1 Semantic Contact Map (SCM)

Following [9], the contact map $C \in \mathbb{R}^{N \times 1}$ represented by heatmap, each $c_i \in C$ ranges between [0, 1], represents the distance to the nearest finger point at each object point. Intuitively, a contact map shows which part of an object might be touched by a hand. However, relying solely on contact maps is insufficient for modeling complex grasps due to ambiguity about how and where the hands touch the object. To overcome this issue caused by the lack of fine-grained contact representations, inspired by [18, 24], we propose the following Semantic Contact Map:

$$SCM_{o \times f} = FingertouchAnalysis(O, H) \in \mathbb{R}^{N \times 5}, \quad (1)$$

where $o$ represents object sampling points $\in R^{2048}$, $f$ represents finger indices $\in R^5$, $O$ represents the object point clouds $\in R^{2048 \times 3}$, $H$ represents the hand point clouds $\in R^{778 \times 3}$.

Different from the part map (one-hot vector whose value is the nearest hand labels) proposed by ContactGen [24] for the entire hand reliant only on the SDF, in our understanding, the contact relationship between fingers and object points can be many-to-many. Moreover, utilizing finger control and emphasizing the finger number rather than the hand part is more convenient and flexible for user-defined controllable generation. Therefore, we heuristically represent finger-object point pairs with a distance below the threshold in SCM as 0. At the same time, Tactile-Guided Constraint designed for SCM is used to better learn embedding conditions. Fig. 3 shows an example of SCM. By processing point clouds of objects and hands, SCM contains information on whether points on

the object are touched and the numbering of the fingers touching those points, which is more important for interactive grasp control. Apart from being generated through Fingertouch-Analysis, the most important aspect of SCM is that it can be customized by the user by clicking and traversing the precalculated weights of the area around the clicked points, hence we can achieve controllable grasp generation.

## 4.2 Dual Generation Framework

Inspired by controllable contact-based method [10, 44], we use conventional contact maps during generation to ensure that the model learns the correct distribution. However, it's challenging to provide realistic contact maps in practical applications and relying solely contact maps introduces ambiguity regarding which part of the hand touches and how it makes contact. Furthermore, past research [15, 18, 24] has shown that single-stage models struggle to generate high-quality grasps. Therefore, we propose a Dual Generation Framework, comprising a Semantic Conditional Module and a Contact Conditional Module. We first use Semantic Contact Map to infer the most probable contact maps based on SCM, then generate controllable grasps under the guidance of Tactile-Guided Constraint. As a result, we achieve controllable generation by starting from a user-defined SCM. Fig. 2 summarizes our process for training and testing.

*4.2.1 Training:* In Semantic Conditional Module, the parameters $C \in \mathbb{R}^{2048}$ are composed of the object's contact map parameters. We use a conditional generation model to infer probable contact maps $C$ based on user-specified or algorithmically predicted Semantic Contact Maps. The process is as follows:

$$\tilde{C}^0 = C\mathcal{G}\mathcal{M}(\tilde{C}^n | SCM, O), \quad (2)$$

where $\tilde{C}^n$ denotes the contact map at noise level n, $O$ denotes the object point clouds and $C\mathcal{G}\mathcal{M}$ denotes conditional generation model. In Contact Conditional Module, MANO parameters [29] $M \in \mathbb{R}^{61}$ are composed of the hand's MANO parameters. We also use a conditional generation model based on contact maps predicted by the Semantic Conditional Module and constrained by Semantic Contact Maps to infer MANO parameters. The training of two modules can be carried out independently. To enable the model to learn additional finger information from simple contact maps, we design Tactile-Guided Constraint, which will be detailed in the next section. The entire process can be represented as:

$$\tilde{M}^0 = C\mathcal{G}\mathcal{M}(\tilde{M}^n | \tilde{C}^0, O), \quad (3)$$

where $\tilde{M}^n$ represents MANO parameters at nosie level n, $O$ represents object point clouds.

*4.2.2 Testing:* As shown in Fig. 2, in the denoising step n, we combine the contact map predicted by the Semantic Conditional Module, along with features extracted by PointNet, as input to the Contact Conditional Module, and estimate $M^0$, which can be represented as:

$$\tilde{C}^{n-1} = C\mathcal{G}\mathcal{M}_{seman}(\tilde{C}^n | SCM, O), \ \tilde{M}^{n-1} = C\mathcal{G}\mathcal{M}_{cont}(\tilde{M}^n | \tilde{C}^0, O),$$
$$\cdots \qquad\qquad\qquad \cdots \qquad (4)$$
$$\tilde{C}^0 = C\mathcal{G}\mathcal{M}_{seman}(\tilde{C}^1 | SCM, O), \qquad \tilde{M}^0 = C\mathcal{G}\mathcal{M}_{cont}(\tilde{M}^1 | \tilde{C}^0, O).$$

## 4.3 Tactile-Guided Constraint (TGC)

In the Contact Conditional Module, simply utilizing contact map as a condition introduces ambiguity regarding which part of the hand is in contact with the object. Based on the Semantic Contact Map, we tailored Tactile-Guided Constraint as follows:

$$\mathcal{L}_{TGC} = \sum_{k=1}^{N} \|O(i_k) - H(j_k)\|_2, \tag{5}$$

where $(i_k, j_k)$ represents $k_{th}$ pair of indices in the SCM that represents touching. $i_k$ denotes the index of points. $j_k$ denotes the index of fingers. $O(i_k)$ denotes coordinates of the object point. $H(j_k)$ denotes coordinates of the centroid for each touched finger. For each finger, we predefine a set of contact points that are weighted differently depending on the distance to the inner surface of the finger. Then a centroid selection algorithm computes the weighted average value from the set to determine the touch centroid for that finger. $N$ denotes the number of contact pairs designated within the SCM.

The Tactile-Guided Constraint loss ($\mathcal{L}_{TGC}$) specifically targets the vertices within the finger sets proximal to the object's surface, ensuring that fingers accurately align with the designated ground-truth contact areas by accurately indexing the point pairs in the SCM and calculating the distance between the centroid of each finger's predefined set of points and the contact point on the object. We adopt the $L_2$ distance to compute the distance error between object and hand point clouds.

We denote the reconstruction loss in the training of Semantic Conditional Module between the predicted contact map and the ground-truth as:

$$\mathcal{L}_R = \|\tilde{C} - C\|_2^2. \tag{6}$$

The reconstruction loss on MANO parameters in the training of Contact Conditional Module is defined in a similar way as:

$$\mathcal{L}_R = \|\tilde{M} - M\|_2^2. \tag{7}$$

At the same time, we design a contact-map-based loss for Semantic Conditional Module training, followed as:

$$B_o = \begin{cases} 0 & \text{if } C_o < \tau_{\text{threshold}}, \\ 1 & \text{otherwise.} \end{cases} \tag{8}$$

$$\mathcal{L}_C = \sum |(1 - B) \odot \tilde{C}|, \tag{9}$$

where $B_o$ represents the binary map's value when the distance $C_o$ between a sampled point on the hand and a corresponding object point $o$ is less than the predefined threshold $\tau_{\text{threshold}}$ and $\odot$ denotes matrix dot product. The efficacy of the $\mathcal{L}_C$ loss function is attributed to its focus on hand-object point pairs, which are systematically selected based on the contact map during the training phase.

Furthermore, to enhance the accuracy of hand posture modeling within Contact Conditional Module, we employ a loss function that measures the discrepancy between predicted vertices $\tilde{V}$ and the ground-truth vertices $V$, which is mathematically represented as:

$$\mathcal{L}_V = \|\tilde{V} - V\|_2^2. \tag{10}$$

In a short summary, the whole loss for training the Semantic Conditional Module is:

$$\mathcal{L}_{semantic} = \lambda_\alpha \cdot \mathcal{L}_R + \lambda_\beta \cdot \mathcal{L}_C. \tag{11}$$

And the training loss of Contact Conditional Module is defined as:

$$\mathcal{L}_{contact} = \lambda_\alpha \cdot \mathcal{L}_R + \lambda_\beta \cdot \mathcal{L}_V + \lambda_\theta \cdot \mathcal{L}_{TGC}. \tag{12}$$

## 5 EXPERIMENT

### 5.1 Experimental Details

*5.1.1* **Datasets:** We employ the GRAB dataset [33] to train the ClickDiff and assess the efficacy of our proposed approach in a unimanual grip scenario. The GRAB [33], which comprises real human grasp data for 51 objects across 10 different subjects, is instrumental in our analysis. Notably, the dataset's test set consists of six objects that are not encountered during training, presenting an opportunity to assess the model's generalization capabilities. Furthermore, ARCTIC [9] is a dataset of dexterously bimanual manipulating articulated objects. It facilitates our exploration into the realm of dexterous bimanual manipulation.

*5.1.2* **Implementation Details:** The ClickDiff is trained on the GRAB [33] and ARCTIC [9] datasets, which takes N = 2048 points sampled from GRAB object surface and N = 600 points sampled from ARCTIC object surface as input. We extract object features using PointNet [4] and adopt the standard Adam optimizer [20] with a learning rate of $1 \times 10^{-5}$ and betas of 0.9 and 0.999. We train our model with batch size of 256 for 600k steps. The loss weights are $\lambda_\alpha$ = 2, $\lambda_\beta$ = 0.5 and $\lambda_\theta$ = 1. The network is implemented in PyTorch and trained on a single NVIDIA RTX 3090 GPU.

*5.1.3* **Evaluation Metrics:** Our objective is to synthesize highly accurate, contact-based, fine-grained hand grips on objects to achieve controllable grasp generation. Thus, we evaluate the evaluation criteria applicable to controllable grasp generation. To evaluate the performance of our model, following [8, 9, 26, 31, 42], we use three metrics previously utilized (MPJPE, MRRPE, CDev) along with a custom metric (Success Rate). To ensure that contacts made by hands touching or holding objects are controllable and accurate, we measure precision using metrics known as Contact Deviation (CDev) and Success Rate. Moreover, capturing the correct posture of the hand and the position of its bones accurately when the hand moves or interacts with an object is crucial. We check these details utilizing Mean Per-Joint Position Error (MPJPE) and Mean Relative-Root Position Error (MRRPE). The definitions are as follows:

- **Mean Per-Joint Position Error (MPJPE):** The $L_2$ distance between the 21 predicted and ground-truth joints for each hand after subtracting its root.
- **Mean Relative-Root Position Error (MRRPE):** Measures the root translation between the hand and object:

$$\text{MRRPE}_{a \to b} = \left\| \left(\mathbf{J}^a - \mathbf{J}^b\right) - \left(\hat{\mathbf{J}}^a - \hat{\mathbf{J}}^b\right) \right\|_2, \tag{13}$$

where $a, b \in \{l, r, o\}$, with $l, r, o$ representing the left hand, right hand, and the object, respectively, $\mathbf{J} \in \mathbb{R}^3$ is the ground-truth root joint position and $\hat{\mathbf{J}}$ is the predicted one. We only adopt this metric on the ARCTIC dataset to measure the bimanual grasps.

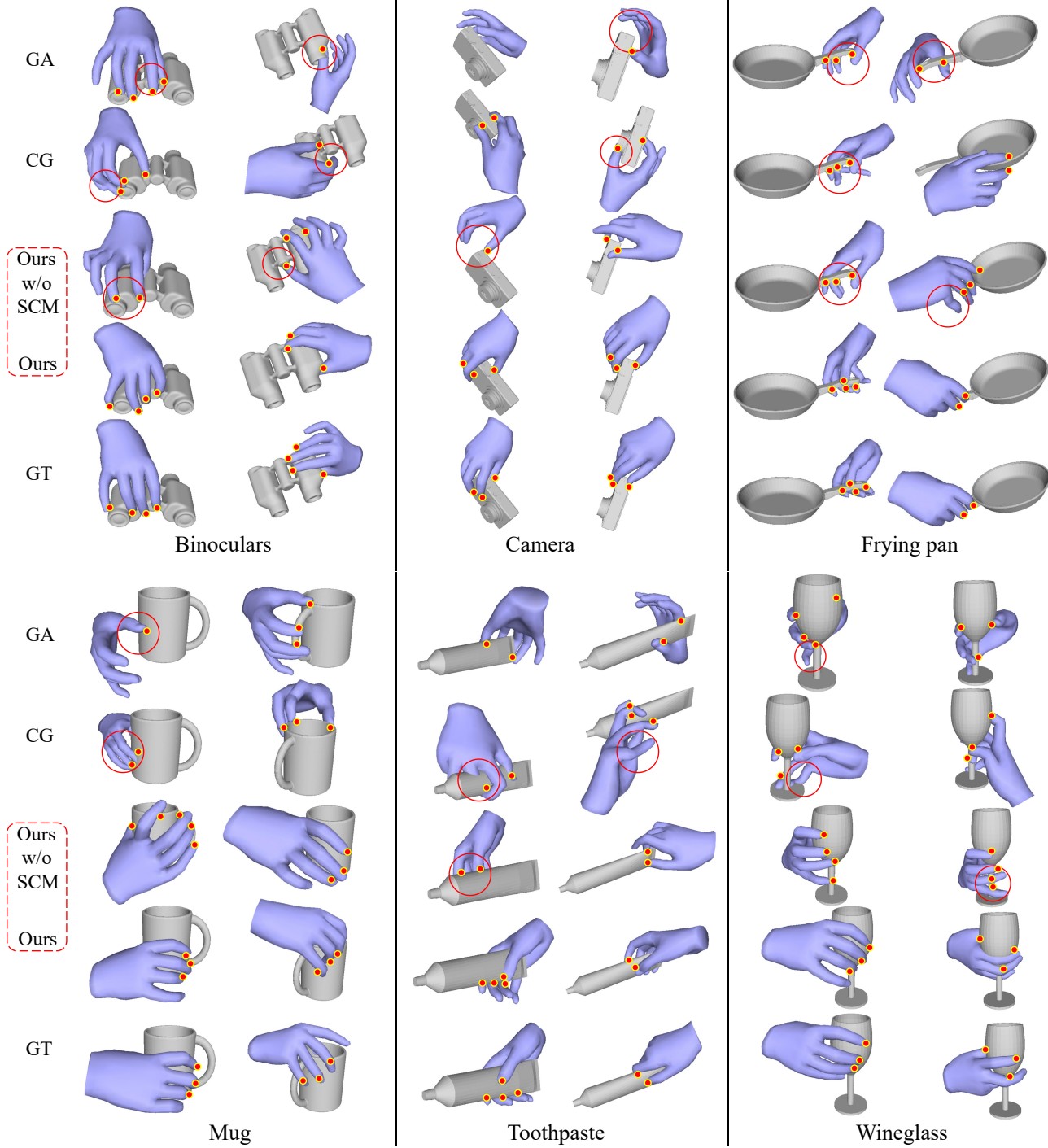

**Figure 4: Qualitative comparison results on GRAB dataset [33]. While GA and CG produce unnatural distortions and huge contact deviations, our method produces more plausible and accurate grasps for unseen objects.**

- **Contact Deviation (CDev):** Measures the deviation between the vertices of the hand and the assumed contact vertices on the object.

$$\frac{1}{C} \sum_{i=1}^{C} ||\hat{\mathbf{h}}_i - \hat{\mathbf{o}}_i||, \qquad (14)$$

where $\{(\mathbf{h}_i, \mathbf{o}_i)\}_{i=1}^{C}$ are $C$ pairs of in-contact hand-object vertices with distance less than 3mm, and $\{(\hat{\mathbf{h}}_i, \hat{\mathbf{o}}_i)\}_{i=1}^{C}$ are the corresponding predictions.

- **Success Rate:** Success rate is defined as the ratio of the size of the union of the point set $A$ in the generated hand point

**Table 1: Quantitative comparison results of state-of-the-art methods and ours on the GRAB dataset [33] on different out-of-domain objects. GN denotes GrabNet [33], GL denotes GOAL [32], GA denotes GraspTTA [15] and CG denotes ContactGen [24].**

| | MPJPE (mm) ↓ | | | | | CDev (mm) ↓ | | | | | Success Rate (%) ↑ | | | | |
|---|---|---|---|---|---|---|---|---|---|---|---|---|---|---|---|
| | GN [33] | GA [15] | GL [32] | CG [24] | | GN [33] | GA [15] | GL [32] | CG [24] | | GN [33] | GA [15] | GL [32] | CG [24] | |
| | ECCV | ICCV | CVPR | ICCV | Ours | ECCV | ICCV | CVPR | ICCV | Ours | ECCV | ICCV | CVPR | ICCV | Ours |
| | 2020 | 2021 | 2022 | 2023 | | 2020 | 2021 | 2022 | 2023 | | 2020 | 2021 | 2022 | 2023 | |
| Binoculars | 87.26 | 64.31 | 80.56 | 82.92 | **40.09** | 103.14 | 78.67 | 87.21 | 98.15 | **43.42** | 46.85 | 63.57 | 46.02 | 33.35 | **75.69** |
| Camera | 79.6 5 | 62.55 | 75.82 | 80.44 | **29.98** | 85.87 | 77.22 | 95.96 | 78.02 | **48.91** | 55.23 | 67.55 | 59.30 | 40.19 | **76.29** |
| Frying pan | 72.17 | 55.12 | 73.76 | 67.31 | **45.40** | 90.44 | **64.28** | 72.41 | 122.83 | 76.35 | 71.60 | **84.17** | 70.77 | 67.54 | 75.73 |
| Mug | 75.11 | 58.67 | 67.12 | 77.78 | **51.53** | 96.47 | 80.21 | 86.83 | 84.24 | **53.26** | 50.03 | 67.79 | 55.88 | 46.54 | **72.45** |
| Toothpaste | 81.83 | 59.82 | 72.95 | 68.62 | **29.34** | 93.72 | 81.34 | 92.73 | 68.72 | **46.33** | 49.56 | **81.34** | 52.43 | 47.57 | 71.54 |
| Wineglass | 88.09 | 64.46 | 83.62 | 83.68 | **44.26** | 95.70 | 98.30 | 85.28 | 75.27 | **46.11** | 58.26 | 58.73 | 61.67 | 53.41 | **68.55** |
| Average | 80.35 | 61.36 | 75.96 | 78.32 | **40.57** | 93.95 | 81.90 | 86.28 | 84.59 | **52.05** | 55.90 | 66.78 | 57.46 | 46.95 | **72.85** |

**Table 2: Quantitative comparison results of state-of-the-art methods and ours on the ARCTIC dataset [9].**

| Method | in/out-of domain | Hand left | Hand right | MPJPE (mm) ↓ | MRRPE (mm) ↓ | CDev (mm) ↓ | Success Rate (%) ↑ |
|---|---|---|---|---|---|---|---|
| GraspTTA [15] | in | ✔ | | 61.57 | 1183.05 | 1174.30 | 53.41 |
| | in | | ✔ | 54.13 | 678.45 | 657.08 | 50.64 |
| | in | ✔ | ✔ | 57.85 | 930.75 | 915.69 | 52.03 |
| Ours (w/o SCM) | in | ✔ | | 48.65 | 75.72 | 77.67 | 78.46 |
| | in | | ✔ | 43.35 | 69.54 | 78.97 | 79.23 |
| | in | ✔ | ✔ | 46.00 | 72.63 | 78.32 | 78.85 |
| Ours (SCM) | in | ✔ | | **39.48** | **66.79** | **70.27** | **81.44** |
| | in | | ✔ | **37.58** | **65.02** | **65.16** | **82.78** |
| | in | ✔ | ✔ | **38.53** | **65.91** | **67.75** | **82.11** |
| GraspTTA [15] | out-of | ✔ | | 51.53 | 876.59 | 1007.14 | 58.89 |
| | out-of | | ✔ | 57.61 | 612.74 | 571.70 | 50.72 |
| | out-of | ✔ | ✔ | 54.57 | 749.15 | 789.42 | 54.81 |
| Ours (w/o SCM) | out-of | ✔ | | 51.78 | 119.20 | 112.35 | 79.16 |
| | out-of | | ✔ | 53.39 | 90.36 | 102.12 | 70.68 |
| | out-of | ✔ | ✔ | 52.59 | 104.78 | 107.24 | 74.92 |
| Ours (SCM) | out-of | ✔ | | **47.16** | **102.54** | **92.06** | **84.01** |
| | out-of | | ✔ | **48.21** | **84.60** | **92.10** | **72.87** |
| | out-of | ✔ | ✔ | **47.69** | **93.57** | **92.08** | **78.44** |

clouds that are in contact with the target object to the size of the point set $B$ in the real hand point cloud that are in contact with the object, relative to the size of set $A$. This metric aims to quantify the contact quality of the generated hand grasps.

## 5.2 Experimental Results

*5.2.1* ***Results on GRAB dataset.*** We evaluate the generalization capability of our models on the GRAB dataset [33]. The dataset's testset, consisting of six objects not previously encountered during training, serves as a benchmark to evaluate our model's adaptability to new scenarios. Tab. 1 shows comparison of our method with GrabNet [33], GOAL [32], ContactGen [24] and GraspTTA [15] on the GRAB dataset [33], our method achieves performance in each metric on almost all objects. Fig. 4 illustrates a significant reduction

in contact deviation when implementing the Semantic Contact Map, revealing the importance of controllable grasp generation. As GraspTTA struggles to produce valid grasps for unseen objects, the contact deviation is substantial while ContactGen often produces unnatural distortions. The results reveal that the hands produced by both GraspTTA and ContactGen lack coordination and deviate significantly from the expected contact areas. In comparison to them, our method achieves notably lower penetration and better stability, which is the closest to ground-truth.

*5.2.2* ***Results on ARCTIC dataset.*** Tab. 2 shows comparison of our method with GraspTTA [15] on the ARCTIC dataset, examining performance on both in-domain and out-of-domain objects. When the unimanual generation method is applied to the bimanual generation, the metric between two hands has a huge gap and incongruous effect, and tends to fail. On the contrary, our approach

**Table 4: Impact of dual framework on the GRAB dataset [33], BM denotes binary map and GM denotes guassian map.**

| Frame | Contact Condtion | | | MPJPE (mm) ↓ | CDev (mm) ↓ | SR (%) ↑ |
|---|---|---|---|---|---|---|
| | BM | GM | SCM | | | |
| Single | ✗ | ✗ | ✗ | 45.92 | 71.07 | 66.34 |
| | ✔ | ✗ | ✗ | 44.74 | 66.28 | 66.96 |
| | ✗ | ✔ | ✗ | 44.81 | 62.02 | 67.88 |
| | ✗ | ✗ | ✔ | 42.54 | 59.91 | 68.23 |
| Dual | ✗ | ✗ | ✔ | **40.57** | **52.05** | **72.85** |

**Table 5: Impact of loss selection on the GRAB dataset [33].**

| Loss | | | MPJPE (mm) ↓ | CDev (mm) ↓ | SR (%) ↑ |
|---|---|---|---|---|---|
| $\mathcal{L}_{TGC}$ | $\mathcal{L}_V$ | $\mathcal{L}_C$ | | | |
| ✗ | ✗ | ✗ | 44.11 | 59.35 | 68.76 |
| ✔ | ✗ | ✗ | 42.02 | 53.24 | 71.87 |
| ✗ | ✔ | ✗ | 40.83 | 60.88 | 68.87 |
| ✗ | ✗ | ✔ | 41.58 | 66.62 | 68.92 |
| ✔ | ✔ | ✔ | **40.57** | **52.05** | **72.85** |

demonstrates superior performance in coordinating both hands, underscoring the efficacy of our dual-frame strategy and the diffusion model's inherent constraints, which facilitate the synchronized generation of bimanual parameters. Our method consistently achieves the lowest joint position deviation and the highest success rate. Notably, the implementation of the Semantic Contact Map further enhances qualitative metrics.

**Table 3: Impact of different contact condition in Semantic Conditional Module on the GRAB dataset [33], BM denotes binary map and GM denotes guassian map.**

| Contact Condtion | | | MPJPE (mm) ↓ | CDev (mm) ↓ | SR (%) ↑ |
|---|---|---|---|---|---|
| BM | GM | SCM | | | |
| ✗ | ✗ | ✗ | 44.87 | 70.92 | 66.35 |
| ✔ | ✗ | ✗ | 46.13 | 68.71 | 67.97 |
| ✗ | ✔ | ✗ | 43.17 | 62.77 | 68.47 |
| ✗ | ✗ | ✔ | **40.57** | **52.05** | **72.85** |

## 5.3 Ablation Study

We first perform ablation studies on GRAB dataset [33] for evaluating the proposed Semantic Contact Map in sec 5.3.1. We then analyse designs of Dual Generation Framework in sec 5.3.2. Finally, we compare different loss selection in sec 5.3.3.

*5.3.1 **Impact of contact condition.*** We compare three different kinds of representations for contact conditions within the Semantic Conditional Module, as summarized in Tab. 3. Initially, the first model employs a binary map that represents whether points on the object are touched. Subsequently, the second model enhances this binary map with a gaussian kernel. Our third model introduces our novel Semantic Contact Map (SCM). Additionally, we examine a model variant devoid of any contact conditions like previous work. The results in the Tab. 3 show that the model utilizing SCM can perform better in both hand posture and contact position, especially with a 18.87mm reduction in Contact Deviation (CDev). The

absence of specific contact information results in a challenging one-to-many mapping problem in grasp prediction. The quantitative results verify the effectiveness of employing the Semantic Contact Map for controllable grasp generation.

*5.3.2 **Impact of dual framework.*** In our assessment of the Dual Generation Framework, we explore five distinct strategies: (1) employing a binary map as the condition in a single-stage generation model to directly synthesize grasps; (2) using a gaussian-kernel-processed binary map as the condition in a single-stage generation model; (3) integrating our SCM within a single-stage generation model; (4) implementing a single-stage generation model without any conditions; and (5) applying SCM in conjunction with a Dual Generation Framework. The experiments presented in Tab. 4 show that the strategy with Dual Generation Framework is very superior quantitatively, as evidenced by an increase of 4.62% in Success Rate and a decrease of 7.86mm in CDev. Previous work [15, 24] has shown that single-stage generation exhibits limitations in accurately generating both hand posture and contact position. This analysis confirms the effectiveness of our Dual Generation Framework, particularly when augmented by SCM.

*5.3.3 **Impact of loss selection.*** We conduct another ablation study to assess the contribution of different losses. The results are shown in Tab. 5. Applying the Tactile-Guided Constraint $\mathcal{L}_{TGC}$ effectively ensures that the fingers align with the designated ground-truth contact regions. Notably, the introduction of $\mathcal{L}_{TGC}$ results in a significant reduction in joint displacement and improvements in contact metrics, exemplified by a 6.11 mm decrease in Contact Deviation (CDev). Experiments demonstrate that our $\mathcal{L}_{TGC}$ constrains the contact position of fingers in the Contact Conditional Module, which solves the contact ambiguity problem well. After adopting the vertice loss $\mathcal{L}_V$, joint and finger related posture metrics, *e.g.*, MPJPE stability decrease. Meanwhile, adding the contact map loss $\mathcal{L}_C$ for Semantic Conditional Module improves contact and displacement related metrics, *e.g.*, CDev and Success Rate, attributable to its focus on hand-object point pairings during training. These results imply that $\mathcal{L}_{TGC}$, $\mathcal{L}_V$ and $\mathcal{L}_C$, in concert, improve the grasp generation, in alignment with the intended design of these loss functions.

*5.3.4 **Impact of controllable generation on time.*** We analyze the influence of the addition of controllable contact conditions on time. In terms of total time, our controllable grasp generation time has increased by only 30% (from 13.5 hours to 17.5 hours) compared to the model devoid of any contact conditions with batch size of 128 for 600k steps on a single NVIDIA RTX 3090 GPU.

## 6 CONCLUSION

In this work, we introduce ClickDiff: Click to Induce Semantic Contact Map for Controllable Grasp Generation with Diffusion Models. To solve the contact ambiguity problem and achieve controllable grasp generation, we propose a simple and efficient Semantic Contact Map that can be defined by users by clicking. At the same time, Dual Generation Framework and Tactile-Guided Constraint are proposed to utilize SCM. Our method demonstrates superior performance to existing grasp generation methods, both qualitatively and quantitatively.

## ACKNOWLEDGMENTS

This work was supported by Shenzhen Innovation in Science and Technology Foundation for The Excellent Youth Scholars (No. RCYX20231211090248064), National Natural Science Foundation of China (No. 62203476).

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
