# OpenReview forum: "ClickDiff: Click to Induce Semantic Contact Map for Controllable Grasp Generation with Diffusion Models"
_acmmm.org/ACMMM/2024/Conference — MM2024 Oral_

### Official Review · Reviewer_djqp · 2024-05-11

**Rating:** 4
**Confidence:** 3

**Summary:**

This work significantly contributes to multimedia/multimodal processing by advancing the state-of-the-art in hand grasping generation through the integration of diffusion models. By generating more natural and accurate hand grasps, this research enhances the interactivity and realism in multimedia applications such as virtual reality (VR), augmented reality (AR), and robotics. Specifically, the controllable grasp generation facilitates more nuanced and realistic human-computer and human-robot interactions, allowing for a more immersive user experience in VR and AR scenarios, and more effective grasp execution in robotics.  Moreover, the use of diffusion models for hand grasp generation opens up new possibilities for synthesizing multimodal data. Thus, this work not only pushes the boundaries of what's possible in hand grasp synthesis but also contributes to the broader field of multimedia/multimodal processing by enhancing the quality of interactive experiences.

**Strengths:**

novelty

**Limitations:**

insufficient evaluation

**Suitability:**

3

---

### Official Review · Reviewer_niWb · 2024-05-24

**Rating:** 6
**Confidence:** 4

**Summary:**

This paper proposes a novel method that generates the hand grasp given the contact points on the mesh by clicking. Previous methods, such as ContactGen and GraspTTA, can generate the physical-plausible "touch", but cannot generation is very random, and cannot make sure the generated grasp is meaningful for the downstream tasks. This method can control the generation with user expectations. Experiments show that the results on GRAB and ARCTIC datasets, further demonstrate the efficacy and robustness of ClickDiff, even with previously unseen objects.

**Strengths:**

1. This is a solid work that tackles a real-world problem.
2. The motivation and paper writing is good.
3. The paper also conducts a thorough comparison with various methods using multiple test datasets and unseen objects.

**Limitations:**

Table 1's comparison is insufficient and somewhat unfair because GraspTTA and ContactGen do not provide contact click points, while our method does. This means our method benefits from having ground truth information in a different format. As a result, this experiment only demonstrates our motivation but does not conclusively show that our method is superior. I suggest comparing the results using the framework from ContactGen or ContactTTA given a click. This comparison will effectively demonstrate that our framework outperforms VAE/GAN-based methods.

**Suitability:**

3

---

### Official Review · Reviewer_qApF · 2024-05-24

**Rating:** 5
**Confidence:** 2

**Summary:**

This paper proposes a diffusion model to generate realistic hand object interactions. It introduces a nvoel representation SCM to reflect the contact between the hand and the object. Based on SCM, it can infer the contact map and finally generate realistic grasps. In the experimental part, the propsed method demonstrates better performance against state-of-the-art methods.

**Strengths:**

- It introduces a novel SCM representation to analyze the contact patterns between the hand and the object.
- The proposed dual learning framework together with tactile-guided constraint allieviate the difficulty of the learning process.
- The generation process is easy to control and shows impressive qualitative results.

**Limitations:**

- Missing failure cases of the proposed method. Though the method shows good performance on unseen object at test time, it would be good to also put some failure cases to help readers know what is the limitation and bottleneck for the proposed method.
- In Table 1, the proposed method does not show consistent advantages over previous methods. Especially for flying pan, the results seem much worse than [16] on Cdev and success rate. What is the reason behind this phenomenon?

**Suitability:**

2

---

### Official Review · Reviewer_ajGV · 2024-06-04

**Rating:** 3
**Confidence:** 3

**Summary:**

This paper proposes a semantic contact map for controllable grasp generation with a diffusion model. Basically, the main idea is to separate the five fingers in the contact map and employ a diffusion model to generate the grasp pose with the condition on the contact map. The writing is easy to follow and the experimental results show some good results.

**Strengths:**

- well-written paper and easy to follow. The main idea is clearly presented.
- The motivation makes sense to me. Using five individual contact maps introduces more semantic information to the generation process.

**Limitations:**

- The experimental results are not sufficient. HOI4D is missing from the evaluation and it is quite well known dataset. It is recommended to involve this dataset. In addition, the training/testing splitting is not clear for GRAB and ARCTIC datasets. This makes the results unconvincing.
- only [16] and [25] are involved for comparison. There are many related works that the authors have shown inspiration in their paper. These works should be involved, e.g. [11,44,19,1,9,10,27,32]. Otherwise, the evaluation result is quite selective.

**Suitability:**

3

---

### Meta-Review · Area_Chair_ao68 · 2024-06-26

**Recommendation:** Accept (Oral)
**Confidence:** 5

**Metareview:**

This paper introduces a novel method for generating controllable hand grasps using diffusion models and a semantic contact map (SCM). The main innovation lies in the separation of the five fingers in the contact map, which allows for more detailed and semantic-rich control over the grasp generation process. By conditioning the diffusion model on this SCM, the method can generate realistic and physically plausible hand-object interactions. The proposed approach shows significant improvements over state-of-the-art methods in terms of the realism and applicability of the generated grasps, as demonstrated through experiments on the GRAB and ARCTIC datasets.

The paper is well-received for its clarity, thorough experimentation, and the introduction of a dual learning framework that eases the learning process. Notably, it enables user-controlled grasp generation through simple interactions with the contact points on the mesh, catering to specific user expectations and enhancing the grasp's relevance for downstream tasks. This method may contribute to multimedia and multimodal processing, particularly in enhancing interactive experiences in VR, AR, and robotics by facilitating more nuanced human-computer and human-robot interactions. All reviewers have agreed to accept the paper.